# The Effects of Bergamot Polyphenolic Fraction, *Cynara cardunculus*, and *Olea europea* L. Extract on Doxorubicin-Induced Cardiotoxicity

**DOI:** 10.3390/nu13072158

**Published:** 2021-06-23

**Authors:** Jessica Maiuolo, Irene Bava, Cristina Carresi, Micaela Gliozzi, Vincenzo Musolino, Federica Scarano, Saverio Nucera, Miriam Scicchitano, Francesca Bosco, Stefano Ruga, Maria Caterina Zito, Francesca Oppedisano, Roberta Macri, Annamaria Tavernese, Rocco Mollace, Vincenzo Mollace

**Affiliations:** 1IRC-FSH Department of Health Sciences, University “Magna Græcia” of Catanzaro, 88100 Catanzaro, Italy; jessicamaiuolo@virgilio.it (J.M.); irenebava@libero.it (I.B.); carresi@unicz.it (C.C.); micaela.gliozzi@gmail.com (M.G.); xabaras3@hotmail.com (V.M.); federicascar87@gmail.com (F.S.); saverio.nucera@hotmail.it (S.N.); miriam.scicchitano@hotmail.it (M.S.); boscofrancesca.bf@libero.it (F.B.); rugast1@gmail.com (S.R.); mariacaterina.zito@libero.it (M.C.Z.); oppedisanof@libero.it (F.O.); robertamacri85@gmail.com (R.M.); an.tavernese@gmail.com (A.T.); mollace.rocco@gmail.com (R.M.); 2Nutramed S.c.a.r.l, Complesso Ninì Barbieri, Roccelletta di Borgia, 88021 Catanzaro, Italy; 3IRCCS San Raffaele, Via di Valcannuta 247, 00133 Rome, Italy

**Keywords:** doxorubicin, rat embryonic cardiomyoblasts, Bergamot Polyphenolic Fraction, oleuropein, *Cynara cardunculus*, endoplasmic reticulum, nutraceutical support

## Abstract

Doxorubicin is an anthracycline that is commonly used as a chemotherapy drug due to its cytotoxic effects. The clinical use of doxorubicin is limited due to its known cardiotoxic effects. Treatment with anthracyclines causes heart failure in 15–17% of patients, resulting in mitochondrial dysfunction, the accumulation of reactive oxygen species, intracellular calcium dysregulation, the deterioration of the cardiomyocyte structure, and apoptotic cell death. Polyphenols have a wide range of beneficial properties, and particular importance is given to Bergamot Polyphenolic Fraction; Oleuropein, one of the main polyphenolic compounds of olive oil; and *Cynara cardunculus* extract. These natural compounds have particular beneficial characteristics, owing to their high polyphenol contents. Among these, their antioxidant and antoproliferative properties are the most important. The aim of this paper was to investigate the effects of these three plant derivatives using an in vitro model of cardiotoxicity induced by the treatment of rat embryonic cardiomyoblasts (H9c2) with doxorubicin. The biological mechanisms involved and the crosstalk existing between the mitochondria and the endoplasmic reticulum were examined. Bergamot Polyphenolic Fraction, Oleuropein, and Cynara cardunculus extract were able to decrease the damage induced by exposure to doxorubicin. In particular, these natural compounds were found to reduce cell mortality and oxidative damage, increase the lipid content, and decrease the concentration of calcium ions that escaped from the endoplasmic reticulum. In addition, the direct involvement of this cellular organelle was demonstrated by silencing the ATF6 arm of the Unfolded Protein Response, which was activated after treatment with doxorubicin.

## 1. Introduction

Doxorubicin (DOXO) is a common chemotherapeutic drug that is widely prescribed in oncology due to its cytotoxic effects [1,2,3,4]. However, the clinical use of DOXO is limited, as its accumulation is associated with progressive myocardial damage, which leads to congestive heart failure [5,6,7,8]. For this reason, patients have often been forced to discontinue DOXO treatment and switch to an alternative agent, which can result in clinical and psychological distress in cases of reduced treatment efficacy [9]. The available scientific data show that the likelihood of heart failure occurring following DOXO treatment is based on the dosage of the drug used, whereby the percentage of patients with cardiac disorders increases as the dose increases from 3% (intake of 400 mg/m^2^) to 17% (intake of 700 mg/m^2^) [10,11]. A careful cellular and molecular analysis showed that DOXO exerts its negative effects on mitochondrial health, predisposing patients to oxidative stress through the accumulation of reactive oxygen species (ROS) [12,13]. In addition, mitochondrial dysfunction induces intracellular calcium dysregulation [14], deterioration in the structure of cardiomyocytes [15], the alteration of heart energy homeostasis [16,17], and apoptotic cell death [18,19]. Therefore, DOXO-induced cardiotoxicity represents a major challenge in the development of treatments capable of reducing its side effects.

Over the last two decades, interest in natural products and their health effects has increased exponentially. Among the natural compounds available, particular importance has been given to bergamot (Citrus bergamia, Risso et Poiteu), which is grown for its countless beneficial pharmacological effects, including specific cardioprotective properties, thanks to its antioxidant, anti-inflammatory, and lipid-lowering effects [20,21,22,23]. In fact, clinical and experimental studies carried out on animal and cellular models have shown that Bergamot Polyphenolic Fraction (BPF) induces hypolipemic and anti-aterogenic effects, whereby it interferes with the autophagic pathway and prevents pathogenic lipid accumulation [24,25,26,27,28,29,30,31].

Olive oil is obtained from the pressing of olives, the fruit of the olive tree (*Olea europaea*), and represents one of the main constituents of the Mediterranean diet. In recent years, oleuropein (OLE), one of the main polyphenolic compounds of olive oil, has attracted the attention of the scientific world due to its countless beneficial health properties [32,33,34], particularly its antioxidant effects [35,36,37,38].

Cynara cardunculus is a fundamental component of the Mediterranean diet, as it is rich in antioxidants, particularly flavonoids and sesquiterpenes [39]. Cynara cardunculus (CYN) extract is a good source of phenolic compounds and has potential therapeutic properties. It has been shown to possess strong antiproliferative, antioxidant [40], antidiabetic, and antimicrobial activities [41], as well as hepato-protective and lipid-lowering effects [42,43,44].

Since it is known that DOXO can cause significant damage to the mitochondria, and that these cellular organelles are linked to the endoplasmic reticulum (ER) through direct crosstalk, we investigated whether the induction of DOXO damage also affects the ER. In particular, ER stress is expressed by activation of the Unfolded Protein Response (UPR), a well-known adaptive response to cellular stress. The UPR is activated by the coordinated actions of three ER transmembrane stress sensors: Inositol-Requiring Enzyme 1α (IRE1α), PKR-like ER kinase (PERK), and activating transcription factor 6 α (ATF6α). Under homeostatic conditions, these three sensors are maintained in an inactive state through their association with Binding Immunoglobulin Protein (BIP); on the contrary, when misfolded proteins accumulate in the ER lumen, BIP dissociates from the ER stress sensors, permitting downstream signaling [45]. Upon BIP release, activation of the three signaling pathways takes place in a different way. PERK and Ire1α are activated by transautophosphoration, while ATF6 is transferred to the Golgi apparatus and transformed into an active transcription factor (p50) [46]. The silencing of a protein through the inactivation of its gene expression allows the pathway in which it is directly involved to be investigated. In fact, the consequences highlighted after silencing a protein provide valuable information about the specific function of the protein of interest. In this study, we investigated the involvement of the ATF6 stress sensors of the UPR following damage induced by DOXO. Under these experimental conditions, silencing the ATF6 protein provided the opportunity to learn more about the role of UPR activation. Following the silencing of the protein, the damage caused by DOXO was more marked. Presumably, the UPR is activated as a protective strategy of the cell as it seeks to recover its original physiological condition following the damage induced by treatment with DOXO [47,48].

The aim of the present study was to investigate the effects of BPF, OLE, and CYN on an in vitro model of cardiotoxicity induced by DOXO treatment. In particular, since the direct involvement of the endoplasmic reticulum following treatment with DOXO has been shown, it was necessary to study the UPR to better understand the role of this pathway in our experimental model.

## 2. Materials and Methods

### 2.1. DOXO and Natural Compounds

Doxorubicin hydrochloride was purchased from Sigma Aldrich.

BPF and CYN were kindly provided by Herbal and Antioxidant Derivatives S.r.l. (Polistena, RC, Italy).

OLE was obtained using a novel sustainable synthetic strategy, as previously described [49].

### 2.2. Cell Cultures

Rat embryonic cardiomyoblasts (H9c2) were supplied by the American Type Culture Collection (Rockville, MD, USA). The cells were cultured in Dulbecco’s modified Eagle’s medium (DMEM) reinforced with 10% fetal bovine serum (FBS), 100 U/mL penicillin, and 100 μg/mL streptomycin in a humidified 5% CO_2_ atmosphere at 37 °C. The medium was changed every 2–3 days and the cells were cultured to 70% confluency in 100 mm dishes. The cells were trypsinized and plated for use in subsequent experiments. Treatment with BPF, OLE, or CYN was performed for 24 h, while DOXO exposure was conducted for 6 or 24 h. For co-treatments, H9c2 cells were pre-treated with BPF, OLE, or CYN for 24 h and subsequently exposed to DOXO for the indicated lengths of time. At the end of the treatment period, the relevant experiments were carried out.

### 2.3. Cell Viability

The MTT assay relies on the observation that live cells with active mitochondria reduce 3-(4,5-dimethylthiazol-2-yl)-2,5-diphenyltetrazolium bromide (MTT) to a visible dark-blue formazan reaction product. This provides an indirect measurement of cell viability, and this was used in the current study. H9c2 cells were plated in 96—well microplates at a density of 6 × 10^3^ and, the next day, were treated with BPF, OLE, or CYN, as indicated. At the end of treatment, the cells were exposed to DOXO for 6 or 24 h. Subsequently, the medium was replaced with phenol red-free medium containing MTT (0.5 mg/mL) and, after 4 h of incubation, 100 μL of 10% SDS was added to each well to solubilize the formazan crystals. The microplates were shaken gently and the optical density was measured at wavelengths of 540 and 690 nm with a spectrophotometer (X MARK Spectrophotometer Microplate Bio-Rad). The results are expressed as percentages of untreated cells and were used to calculate the relative cell viability. To confirm these results, the cytotoxicity was also calculated using the trypan blue dye exclusion test. Trypan blue is a dye used to quantify live cells by labelling only dead cells exclusively. Live cells have an intact cell membrane, and trypan blue cannot penetrate the cell membrane and enter the cytoplasm. On the contrary, the dead cells are characterized by pores on the plasma membrane, and trypan can therefore enter the cytoplasm. Under light microscopy analysis, only dead cells appear blue. The viable and dead cells were counted using a hemocytometer after trypan blue (0.4% *w*/*v*) staining [50]. Cell death was reported as the percentage of stained (non-viable) cells of the total cells counted. The results obtained with both methods showed the same trend (data not shown).

### 2.4. Intracellular ROS Detection

TROS identification was based on the oxidation of the permeable non-fluorescent probe H_2_DCF-DA. H_2_DCF-DA readily diffuses into cells, where intracellular esterases cleave the acetate group of H_2_DCF-DA from the molecule to yield H_2_DCF, which is trapped inside the cells. Intracellular ROS oxidize H_2_DCF to form the highly fluorescent compound DCF. H9c2 cells were plated in 96—well microplates at a density of 6 × 10^4^ and, the next day, were treated as described. At the end of the treatment period, the growth medium was replaced with fresh medium containing H_2_DCF-DA (25 μM). After 30 min at 37 °C, the cells were washed twice to remove the extracellular H_2_DCF-DA and centrifuged. They were then resuspended in PBS and, in the presence (or not) of H_2_O_2_ (100 μM, 30 min of incubation), the fluorescence was evaluated by flow cytometric analysis, using a FACS Accury laser flow cytometer (Becton Dickinson). The results were expressed by setting the control equal to 1 and reporting all other values. H_2_O_2_ was used as a positive control.

### 2.5. Cell Lysis and Immunoblot Analysis

Cell monolayers in 100 mm plates were washed with ice-cold PBS and lysed with pre-heated (to 80 °C) lysis buffer containing 50 mM of Tris-HCl at pH 6.8, 2% SDS, and a protease inhibitor mixture, and immediately boiled for 2 min. The protein concentration in cell lysates was determined by the DCA protein assay, and after the addition of 0.05% bromphenol blue, 10% glycerol, and 2% β-mercaptoethanol, samples were boiled again and loaded in SDS-polyacrylamide gels (8–12%). After electrophoresis, polypeptides were transferred to nitrocellulose filters and blocked with TTBS/milk (TBS 1%, Tween 20, and non-fat dry milk 5%), and then antibodies were used to reveal the respective antigens. Primary antibodies were incubated overnight at 4 °C, followed by incubation with a horseradish peroxidase-conjugated secondary antibody for 1 h at room temperature. Blots were developed using the chemiluminescence procedure. The following primary antibodies were used: a mouse monoclonal anti ATF-6 antibody (BioAcademia) at 1:500 dilution, and a mouse monoclonal anti-actin antibody (Sigma Aldrich) at 1:2000 dilution. Horseradish goat anti-mouse HRP-conjugated antibody was used as a secondary antibody at 1:5000 dilution. The blots were developed with ECL-PRIME reagent (Perkin Elmer, Monza, Italy).

### 2.6. Lipid Extraction: Spectrometric and Flow Cytometric Quantification

Nile Red is a colouring fluorescent compound (excitation/emission maxima ~552/636) that tightly binds to cell lipids and, consequently, has been used to quantify these molecules in cells. Prior to lipid extraction, an analysis of the cell DNA content for each sample was carried out, so the results obtained for the lipid content could subsequently be normalised. This was done to prevent potential lipid quantity modulations from having different amounts of cells. For DNA quantification, we measured nucleic acids fluorometrically using fluorogenic dyes that bind selectively to DNA (ThermoFischer Scientific, Waltham, MA, USA). Concentrations of DNA were measured using the fluorescence signal of the sample, and a calibration curve was generated from standard samples of a known concentration [51]. To analyze the lipid content, the cells were first incubated with 1 µg/mL of Nile Red dye for 15 min and, for the spectrometric assessment, lipid extraction was performed by the Bligh and Dyer method [52], with precise solvent ratios used. Solvent systems should extract lipids without the introduction of non-lipid materials (e.g., sugars, peptides, amino acids, and other water-soluble compounds) and should also fully dissociate and deactivate any lipolytic enzymes. Briefly, a mixture of CHCl3:MeOH:H2O (ratio of 2:2:1.8, *v*/*v*/*v*) was added to each sample following vortexing, and then samples were incubated on ice for 30 min. Samples were centrifuged at 2000 rpm for 5 min at 4 °C. The lower phase (organic) layer was transferred to a new tube. The aqueous layer was re-extracted with 1 mL of 1:1 *v*/*v* chloroform/methanol. Subsequent tests were carried out on the organic phase. Then, the lipid content of each sample was subjected to fluorimetric reading, and achieved results were interpolated with a straight line built with increasing concentrations of Nile Red. Fluorimetric reading was carried out using a VICTOR 2 spectrofluorimeter (Perkin Elmer, Milan, Italy) at the right wavelengths. The results obtained from the spectrofluorimetric reading are expressed as relative fluorescence units (RFU). The value of the untreated cells (control) value was taken as 1 and all other values were related to the control.

For the cytometric analysis, the cells were treated as described and incubated with 1 µg/mL Nile Red dye for 15 min. Later, the cells were washed in PBS (pH = 7.4), dissolved in dimethylsulfoxide (DMSO), and immediately read by the FACS Accury flow cytometer (Becton Dickinson, 20161 Milan, Italy) [53]. The level of fluorescence obtained from each cytometric reading, expressed as a percentage, was quantified using an arbitrarily designed marker. The value of the untreated cells was taken as 1 and all other values were related to the control.

### 2.7. Intracellular Calcium Measurements

The Rhodamine 2 indicator (Rhod 2, Molecular Probes) was used to measure intracellular calcium, due to this indicator’s extensive fluorescence (λ = 581 nm) when it binds to the ion following excitation at λ = 552 nm. The experiments were conducted with a medium free of calcium and magnesium in order to avoid changes in calcium concentrations following the entry of ions from outside the cell. The cell line was treated with DOXO or DOXO+ BPF, OLE, or CYN, as previously described. At the end of the treatment, the cells were exposed to Rhod 2 (5 μM, for 1 h) at 25 °C and were protected from light. Cells were washed and immediately subjected to flow cytometric analysis using a solution free of calcium and magnesium. After the first reading, which provided the basal concentration of the calcium ion, the cells were exposed to thapsigargin (TAPSI, 1 µM for 200 s), which increased the cytosolic calcium concentration by releasing the ion from the endoplasmic reticulum. Under these conditions, three new flow cytometric readings were carried out and, after 100 s, 500 µM of ethylene glycol-bis (β-aminoethyl ether)-N,N,N0,N0-tetraacetic acid (EGTA) was added to chelate calcium. It is important to remember that, before the readings, the cells were also treated with the decoupler of mitochondrial oxidative phosphorylation, carbonyl cyanide-4-(trifluoromethoxy) phenylhydrazone (FCCP, 1 µM), as well as the inhibitor of the mitochondrial ATPase, oligomycin, (1 µg/mL). Both substances excluded the involvement of mitochondrial calcium [54]. All flow cytometric detections were carried out with a FACS Accury apparatus (Becton Dickinson) [55].

### 2.8. ATF6α Silencing

In order to understand the role of ATF6 activation in our experimental model, we decided to silence this protein and assess the consequences. Cells at ~40% confluence were transfected with RNAi duplexes with the use of the RNAiMAX Lipofectamine reagent (Invitrogen, Waltham, MA, USA). ATF6α was silenced with duplex siRNAs at a concentration of 10 nM (Stealth RNAi HSS177036, Invitrogen). Parallel cultures were transfected with equal concentrations of Stealth Negative Universal Control Medium (Invitrogen) [56]. Six hours after transfection, the medium was replaced, and the following day, the cells were treated with DOXO or DOXO + BPF, OLE, or CYN, as indicated. At the end of the treatment period, cells were trypsinized, and the expression of ATF-6 was evaluated.

### 2.9. Annexin V Staining

The cells were treated as indicated above. After the treatment and co-treatment periods, the cells were detached by trypsin, washed twice with cold PBS, and then re-suspended in 1× Binding Buffer (Metabolic Activity/AnnexinV/Dead Cell Apoptosis Kit) at a concentration of 1 × 10^6^ cells/mL. One hundred microliters of the suspension was transferred to a 5 mL culture tube, and 5 μL of FITC Annexin V (BD Biosciences, San Jose, CA, USA) was added. The samples were gently vortexed and incubated for 15 min at 25 °C in the dark. Finally, 400 μL of 1× Binding Buffer and 5 μL of propidium iodide (PI) were added to each tube, and the samples were analyzed by flow cytometry for 1 h. The fluorescence was evaluated by flow cytometric analysis using an Accury FACS laser flow cytometer (Becton Dickinson).

### 2.10. Statistical Analysis

Data are expressed as the mean ± standard deviation, and were statistically evaluated for differences using a one-way analysis of variance (ANOVA), followed by the Tukey–Kramer multiple comparison test (GraphPad software for science).

## 3. Results

### 3.1. Assessment of Cell Viability

H9c2 cells were treated with BPF, OLE, or CYN at different doses for 24 h. As shown in Figure 1 (panels a, b, and c), no reduction in cell viability was observed at any of the concentrations of BPF, OLE, or CYN considered. In light of this result, the most non-toxic and potentially cytoprotective doses of BPF, OLE, or CYN were tested to determine their protection against DOXO.

The concentration of DOXO used and the treatment time points were chosen based on the literature data. The results showed a statistically significant reduction in cell viability after both 6 and 24 h of treatment with 1 μM DOXO (*p* < 0.05; *p* < 0.01 respectively; Figure 1, panels d and e). Interestingly, statistically significant protection against DOXO damage was observed when cells were pre-treated for 24 h with 5 μg/mL of BPF, 1 μM of OLE, or 1 μM of CYN. In particular, BPF, OLE, and CYN increased cell viability when compared with DOXO-induced damage, and this effect was more evident after 6 h of DOXO treatment when compared with 24 h of treatment. In fact, BPF, OLE, and CYN showed equal levels of protection against damage caused by 6 h of DOXO treatment (*p* < 0.05 vs. DOXO; Figure 1, panel d). On the contrary, the damage induced by DOXO for 24 h was only prevented by BPF (*p* < 0.05 vs. DOXO; Figure 1, panel e).

### 3.2. Oxidative Damage

Based on our first data observations, we chose to continue the analysis of the early toxic effects of DOXO by deepening our study of the 6 h treatment of H9c2 cells with DOXO to understand whether the natural compounds considered could provide cellular protection in the early stages of anthracycline damage. Since DOXO causes oxidative damage in cells, the rise of ROS production was quantified in our experimental model. As shown in Figure 2, treatment with DOXO resulted in an increase in ROS production (*p* < 0.01 vs. control). Interestingly, after pretreatment of the cells with BPF, OLE, or CYN for 24 h, statistically significant protection against DOXO-induced ROS accumulation was registered (*p* < 0.05; Figure 2, panel b). Hydrogen peroxide (H_2_O_2_) was used as a positive control. Ten thousand cells per sample were acquired using a FACS Accury laser flow cytometer (Becton Dickinson). No ROS accumulation was observed in cells treated with natural compounds alone (Figure 2, panel b).

### 3.3. Involvement of the Endoplasmic Reticulum

Since the accumulation of ROS and mitochondrial oxidative damage are frequently related to dysfunction of the endoplasmic reticulum, we decided to investigate whether activation of the UPR, a well-known contributor to the restoration of cellular homeostasis under stressful conditions, occurred following treatment with DOXO. The results obtained for our experimental model show that the treatment with DOXO triggered the activation of the ATF6 arm of the UPR. Indeed, a modulation of the cytosolic fraction was observed. This was characterized by a marked and significant reduction in the cytoplasmic ATF6 (c-ATF6) expression level (*p* < 0.001 vs. control; Figure 3, panel b on the left). At the same time, an increase in the nuclear fraction (n-ATF6) was identified (*p* < 0.001 vs. control; Figure 3, panel b on the right). Treatment with natural compounds alone did not affect ATF6 expression, while the pre-treatment of cells with BPF, OLE, or CYN led to a marked recovery of c-ATF6 to a level similar to that of the control (*p* < 0.01 vs. DOXO). This recovery was also highlighted by the simultaneous reduction of n-ATF6 (*p* < 0.001 vs. DOXO). In contrast, the other two arms of the UPR, involving the IRE-1α and PERK transducers, were not involved (data not shown).

### 3.4. Measurement of the Lipid Content

Following the investigation of the DOXO-induced involvement of the endoplasmic reticulum, we decided to evaluate two aspects controlled by this cellular organelle: lipid synthesis and the regulation of calcium ions.

When H9c2 cells were exposed to DOXO treatment, a significant reduction in lipid content was found (*p* < 0.01 vs. control). Conversely, in cells pre-treated with BPF, OLE, or CYN, DOXO-induced lipid reduction was prevented, bringing the values similar to those of the control (*p* < 0.05 vs. DOXO). Treatment with natural compounds alone did not result in any change in the lipid content compared with the control (Figure 4). The results obtained from the spectrofluorimetric reading are expressed as relative fluorescence units (RFU). Then, the untreated cells (control) were given a value of 1, and the values of all other cells were related to this.

### 3.5. Modulation of Calcium Ions

In our experimental model, the direct involvement of the endoplasmic reticulum led us to assess variations in the concentration of calcium ions following the treatments. Our data show that treatment with DOXO resulted in an increase in basal calcium ions as compared with the control (*p* < 0.01). This is compatible with the idea that calcium ions escape from the endoplasmatic reticulum under such conditions (Figure 5). Pre-treatment with BPF, OLE, or CYN reduced basal calcium levels to levels similar to that of the control (Figure 5). In these experiments, the protective effect of OLE was lower than those exerted by BPF and CYN, and basal calcium levels similar to those found in cells treated with DOXO alone were shown (Figure 5). On the contrary, pre-treatment with BPF or CYN provided significant protection against the effects of DOXO (*p* < 0.05).

To demonstrate the reliability of the system, TAPSI was added to the samples in order to provoke the release of calcium ions from the endoplasmic reticulum and to highlight the increase in ions detected. Then, we used the chelating agent EGTA to lower the calcium ion levels. This experimental protocol assured us that calcium ions from outside the cell and calcium ions from mitochondria were both excluded; therefore, the detected calcium ions corresponded to real cytosolic calcium ions, and oscillations could only be caused by the leaking of ions from the endoplasmic reticulum. A further test of the functionality of this experimental protocol was provided by the positive control reading. In this case, the cells were initially treated with TAPSI at a concentration of 1 μM for 5 min, resulting in the release of calcium ions from the endoplasmic reticulum. Subsequently, when TAPSI was re-added to the samples at the indicated doses and times, an additional increase in calcium ions occurred but at very low concentrations, which demonstrated that most of the ions had already escaped from the endoplasmic reticulum.

### 3.6. Silencing of ATF6 and Its Consequences

In order to understand the role of ATF6 in our experimental model, the silencing of this protein was performed. First, an assessment of the expression level of ATF6 was carried out in both silenced and non-silenced samples in order to evaluate whether silencing had occurred properly. A difference in protein expression between the non-silenced sample (N.S.) and the silenced sample (S) was found; the difference was statistically significant (*p* < 0.001 vs. N.S.) and is shown in Figure 6 (panel a).

As a result of ATF6 silencing, cell damage induced by DOXO treatment increased significantly, showing a high level of apoptotic cell death. Indeed, an increase in secondary apoptotic death occurred compared with the control (*p* < 0.001), as evidenced by the Annexin V/PI positive cells in Figure 6 (panel b). This result shows that activation of the ATF6 arm of the UPR protects against damage induced by DOXO, since its silencing resulted in a worsening of DOXO’s effects. This interpretation was also demonstrated by the results obtained after the pre-treatment with natural compounds, which only partially succeeded in alleviating the substantial level of damage induced by DOXO (*p* < 0.05). Further, Annexin V positive /PI positive cells were still found to be present. Staurosporine (STAURO), an apoptosis inducer and a potent and non-selective inhibitor of protein kinases, was used as a positive control. The relative quantification results are shown in Figure 6 (panel c).

The lipid content was analyzed after ATF6 silencing by spectrophotometric or fluorimetric analyses. Interestingly, the results showed that the silencing of ATF6 led to a greater reduction in the lipid content than that observed under the same experimental conditions in non-silenced cells following DOXO treatment (*p* < 0.001; Figure 7, panel b). When ATF6 was silenced and cells were pretreated with natural compounds and exposed to DOXO, a partial inversion of the lipid reduction effect was observed, with only a small increase (*p* < 0.05) compared with the cells that were silenced and treated with anthracycline noted, as can be seen from the comparison of Figure 7 (panel b) with Figure 4. Therefore, activation of the ATF6 arm of the UPR was found to have a protective role against damage induced by DOXO.

Finally, the basal cytoplasmic calcium ion level was measured after ATF6 silencing. As shown in Figure 8, ATF6 silencing led to a significant modulation of basal calcium ion levels. In particular, the treatment of cells with DOXO caused a greater release of calcium ions compared to that obtained under the same experimental conditions in non-silenced cells (*p* < 0.01). Intriguingly, under these experimental conditions, pre-treatment with BPF, OLE, or CYN failed to prevent the detrimental effects induced by DOXO (Figure 8, panels a and b). To better appreciate the differences in calcium modulation in cells silenced for ATF6 compared with non-silenced cells, Figure 8 and Figure 5 can be compared.

## 4. Discussion

The main purpose of this paper was to evaluate the protective role of natural compounds (BPF, OLE, and CYN) against DOXO-induced damage in embryonic rat cardiomyoblasts. H9c2 cells were cultured and treated with DOXO at a concentration of 1 μM for 6 and 24 h. In both cases, damage was induced by DOXO, as evidenced by a reduction in cell viability (*p* < 0.05; *p* < 0.01 respectively). The protection given by BPF, OLE, and CYN depended on the duration of DOXO exposure and, therefore, on the reversibility of the damage caused. While damage caused by DOXO for 6 h was prevented by all three natural compounds considered (*p* < 0.05), DOXO treatment for 24 h caused greater damage, which only BPF could partially diminish. Most likely, treatment with DOXO for 24 h was responsible for significant damage which OLE and CYN could not resolve. Indeed, it is known that the defence of cellular integrity by natural compounds depends on the extent of the existing damage, and damage exceeding a certain limit can no longer be restored [57,58]. In light of the initially observed data, we decided to continue our research study by deepening the analysis through the treatment of H9c2 cells with DOXO for 6 h to prove that damage can be repaired when it is “caught in time”, and that the natural compounds chosen may be valuable aids in this process. Our subsequent experiments clearly demonstrated that treatment with DOXO for 6 h resulted in oxidative damage, as detected by the detrimental accumulation of ROS (*p* < 0.01 vs. control). This damage was significantly reduced by BPF, OLE, and CYN (*p* < 0.05). Again, BPF was found to be the most active compound, while OLE and CYN had smaller effects but with the same magnitude. Since the oxidative damage demonstrated in our experimental system is also frequently associated with endoplasmic reticulum dysfunction [59,60,61], we aimed to assess whether DOXO treatment is responsible for this dysfunction or not. The results obtained showed that treatment with DOXO in the early stages of the onset of damage activated the ATF6 arm of the UPR, leading to an altered lipid content and the dysregulation of calcium ions, thus highlighting the occurrence of endoplasmic reticulum dysfunction [62]. These data are in agreement with those contained in the scientific literature, which show that there is crosstalk between the endoplasmic reticulum and mitochondria [63,64,65]. Activation of the ATF6 arm was demonstrated through a reduction in its cytoplasmic fraction (*p* < 0.001 vs. control) and a concomitant increase in the nuclear fraction (*p* < 0.001 vs. control) following treatment with DOXO for 6 h. Interestingly, pre-treatment with BPF, OLE, or CYN led to a complete or partial shutdown of this pathway. This result suggests that the UPR may be an attempt by the endoplasmic reticulum to reduce stress and restore cellular homeostasis in the early stages of cytotoxic damage [66,67,68]. The involvement of the endoplasmic reticulum was also demonstrated by a reduction in the lipid content and by the alteration of cytosolic calcium levels. Indeed, the synthesis of lipids necessary for the cell and the regulation of calcium ion release are two important functions performed by the endoplasmic reticulum [69,70,71,72]. When the cells were co-treated with both DOXO and natural compounds, the lipid content increased (*p* < 0.05 vs. DOXO) and the basal calcium ion level reduced, resulting in values similar to those detected in untreated cells (*p* < 0.05). Thus, once again, BPF, OLE, and CYN managed to reduce the early damage caused by DOXO.

The recent scientific literature reports that activation of the UPR is closely connected with the cellular autophagic protective response [73], which leads to reductions in apoptotic death [74] and carcinogenic progression [75]. Therefore, it is likely that shutdown of the UPR coincides with a dangerous cellular situation under stressful conditions. In our experimental model, ATF6 silencing resulted in a worsening of the side effects induced by DOXO when compared with non-silenced cells under the same experimental conditions. The damage was demonstrated by a further disproportionate increase in cell mortality (*p* < 0.001), an intense reduction in the lipid content (*p* < 0.001), and a massive level of dysfunction in the release of calcium ions from the endoplasmic reticulum (*p* < 0.01). These results show that UPR activation represents a defensive mechanism that allows cells to react to and overcome DOXO-induced damage. In fact, when this pathway undergoes artificial knockdown, DOXO-induced damage is more severe. It is important to point out that, under these more severe conditions, BPF, OLE, and CYN can only reduce DOXO-induced damage slightly. Because the natural compounds were shown to revert and/or significantly avoid the damage caused by DOXO when there was no ATF6 silencing, the hypothesis regarding the protective role of the UPR on the endoplasmic reticulum is supported. In light of the results obtained, this study has shown some very interesting aspects which should be discussed further. First of all, DOXO causes a significant level of dysfunction in embryonic cardiomyoblasts, which becomes more severe and is potentially irreversible after longer (24 h), but not shorter exposure times (6 h) [76]. This interpretation of the results is justified by knowledge of the mechanisms through which doxorubicin exerts its proapoptotic effects. This anthracycline is able to intercalate into the double helix of DNA, inhibiting the enzyme topoisomerase II and altering its stability [77]. DOXO is only able to induce cell death by apoptosis at particular doses and under particular treatment conditions. It is known that damage caused by DOXO is dependent on the dose used [78] and the continuity of exposure; a number of previous in vitro studies have reported that continuous treatment with anthracycline is more harmful than discontinuous treatment involving occasional interruptions to the presence of DOXO [79].

Another important point is that cells presumably attempt to recover homeostasis by activating an immediate endoplasmic reticulum response. Activation of the UPR could be a defense strategy for cells to overcome stressful conditions [80,81]. This interpretation was demonstrated by the silencing of ATF6, as shutdown of UPR activation led to a drastic increase in DOXO-induced damage. A potentially protective role of the UPR has been widely reported in the scientific literature [82,83,84,85].

Finally, we have demonstrated, for the first time, that natural compounds of interest (BPF, OLE and CYN) are able to protect H9c2 cells from the damage induced by treatment with DOXO. In fact, the reduction of cell viability, the substantial oxidative damage, the reduction of the lipid content, and the alteration of the concentration of the calcium ion (all damages induced by exposure to DOXO) have been reverted or positively modulated by BPF, OLE and CYN in previous studies [86]. Consequently, these natural compounds may provide nutraceutical support to pharmacological therapies involving anthracyclines, support which serves to reduce and/or avoid the onset of cardiac damage and allows anthracyclines to play their pharmacological role. Together with the addition of BPF, OLE, and CYN, the activation of the UPR of the endoplasmic reticulum could play an important role in resolving the toxicity induced by pharmacological treatment with DOXO.

## 5. Future Perspectives

The protection afforded by pre-treatment with the natural compounds of interest, following the early damage induced by DOXO, provided valuable information regarding the potential use of these substances along with chemotherapy treatment with anthracycline. Since the cellular mechanism with which BPF, OLE, and CYN are able to exert their protective effects in this experimental model is presumably to be found in their common high number of polyphenols, it would be interesting to identify the components most represented in these extracts in order to test them individually. Therefore, the continuation of this work could follow two important phases: first of all, chemical analyses carried out with HPLC could provide details of the composition of the natural compounds of interest; second of all, further experiments should be carried out using the components found and more widely represented in BPF, OLE, and CYN. By following this direction, we may be able to understand if a single component or several components are responsible for the effects generated and, at the same time, we can provide important suggestions for pre-clinical and clinical medicine.

## Figures and Tables

**Figure 1 nutrients-13-02158-f001:**
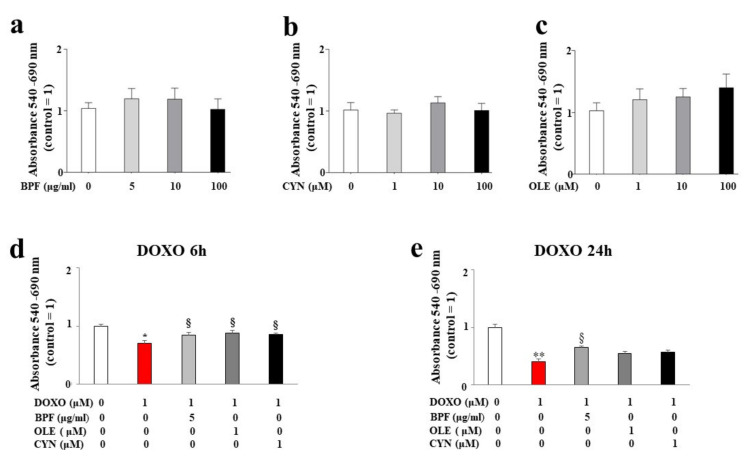
Measurement of cell viability. Cell viability following treatment with different concentrations of BPF (panel **a**), OLE (panel **b**), and CYN (panel **c**) was determined. The doses of the natural compounds were chosen arbitrarily and did not result in a reduction in cell viability compared with control cells. Cell viability after 24 h of pre-treatment with the lowest concentrations of BPF (5 μg/mL), OLE (1 μM), or CYN (1 μM), and subsequent treatment with 1 μM of DOXO for 6 (**d**) or 24 h (**e**) was assessed. The values of three independent experiments are expressed as the means ± standard deviations. * denotes *p* < 0.05 vs. control; ** denotes *p* < 0.01 vs. control; § denotes *p* < 0.05 vs. DOXO. Analysis of Variance (ANOVA) was followed by the Tukey–Kramer comparison test.

**Figure 2 nutrients-13-02158-f002:**
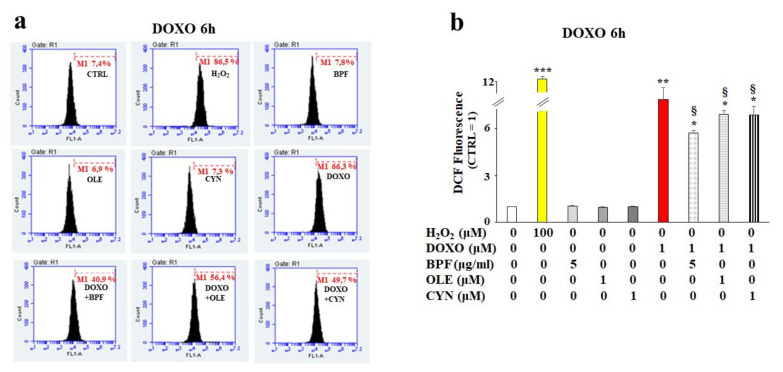
ROS accumulation measurement. H9c2 cells were treated with 1 μM DOXO for 6 h or pre-treated with natural compounds for 24 h and then exposed to DOXO for 6 h. At the end of the treatment period, ROS accumulation was assessed by cytometric analysis. A right shift of the cell population indicates increases in fluorescence and ROS production compared with the control, while a left shift indicates reductions in fluorescence and ROS. A representative cytometric analysis of three independent experimental sets was performed (panel **a**). In each plot, M1 represents an arbitrarily designed marker that can be used to examine fluorescence variations. 10,000 cells per sample were acquired using a FACS Accury laser flow cytometer (Becton Dickinson). The histogram presents the results of the quantification of the three independent experimental sets (panel **b**). Relative quantification was conducted by setting the value of the control to 1 and comparing all other values. H_2_O_2_ (100 μM for 30 min) was used as a positive control. The values of three independent experiments are expressed as the means ± standard deviations. * denotes *p* < 0.05 vs. control; ** denotes *p* < 0.01 vs. control; *** denotes *p* < 0.001 vs. control. § denotes *p* < 0.05 vs. DOXO. The Analysis of Variance (ANOVA) was followed by the Tukey–Kramer comparisons test.

**Figure 3 nutrients-13-02158-f003:**
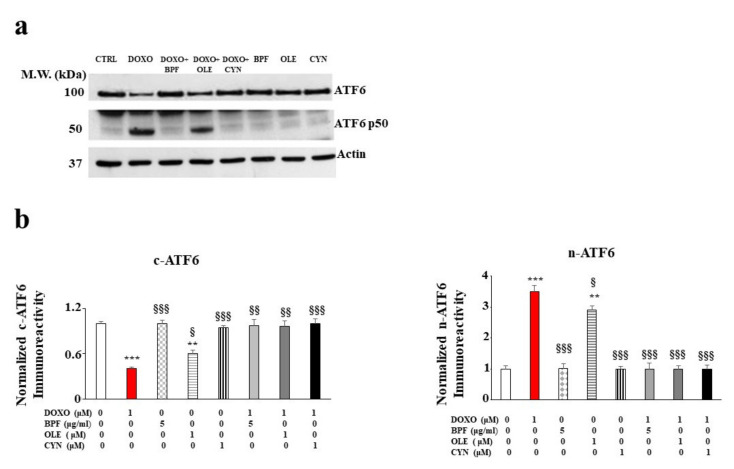
The role of ATF6 UPR arm activation. A representative Western blot of c-ATF6 and n-ATF6 modulation after pre-treatment with BPF (5 μg/mL), OLE (1 μM), or CYN (1 μM) for 24 h and treatment with 1 μM DOXO for 6 h is shown in panel **a**). A graph showing the protein expression levels of c-ATF6 and n-ATF6 is presented in panel **b**). The values of three independent experiments are expressed as the means ± standard deviations. ** denotes *p* < 0.01 vs. control; *** denotes *p* < 0.001 vs. control. § denotes *p* < 0.05 vs. DOXO; §§ denotes *p* < 0.01 vs. DOXO; §§§ denotes *p* < 0.001 vs. DOXO. The Analysis of Variance (ANOVA) was followed by the Tukey–Kramer comparisons test.

**Figure 4 nutrients-13-02158-f004:**
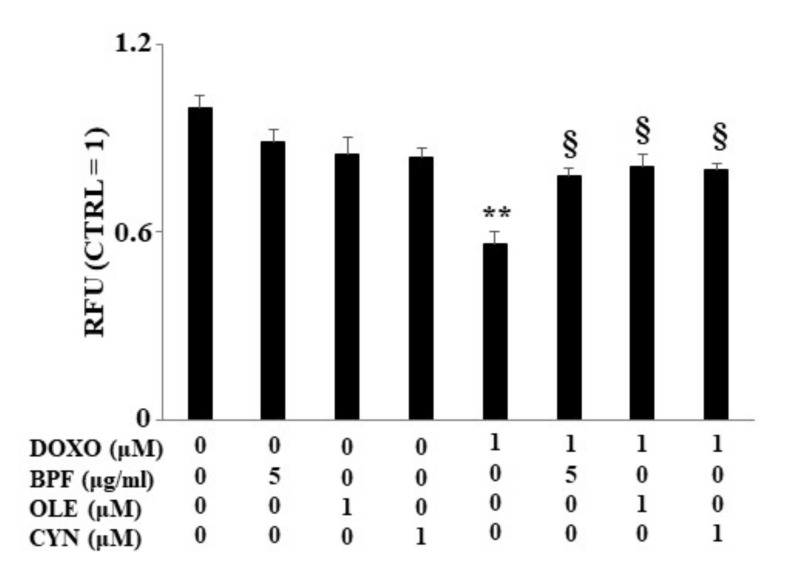
Lipid content measurement. The lipid content was analyzed spectrophotometrically after treatment with 1 μM DOXO for 6 h; BPF (5 μg/mL), OLE (1 μM), or CYN (1 μM) for 24 h; or co-treatments. Values of three independent experiments are expressed as the means ± standard deviations. ** denotes *p* < 0.01 vs. control; § denotes *p* < 0.05 vs. DOXO. The Analysis of Variance (ANOVA) was followed by the Tukey–Kramer comparison test.

**Figure 5 nutrients-13-02158-f005:**
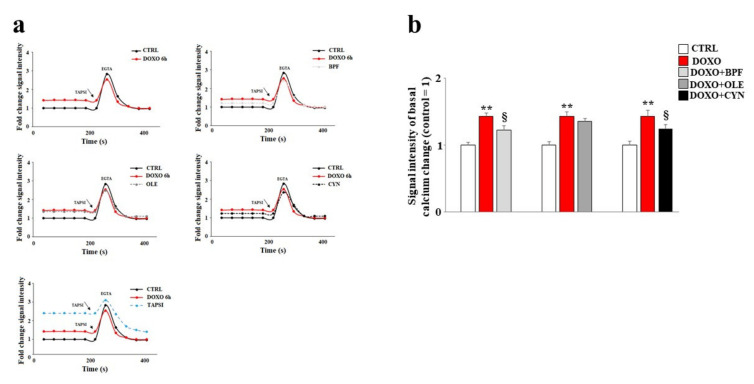
Measurement of basal calcium ion levels. Modulation of the basal concentration of calcium ions released from the endoplasmic reticulum after treatment with 1 μM of DOXO alone or after co-treatment with DOXO and BPF (5 μg/mL), OLE (1 μM), or CYN (1 μM) was measured (panel **a**). Treatment with TAPSI was used as a positive control. Relative quantification of the signal intensity of the basal calcium change was conducted (panel **b**). Representative graphs of three independent experiments are shown. ** denotes *p* < 0.01 vs. control; § denotes *p* < 0.05 vs. DOXO. The Analysis of Variance (ANOVA) was followed by the Tukey–Kramer comparison test.

**Figure 6 nutrients-13-02158-f006:**
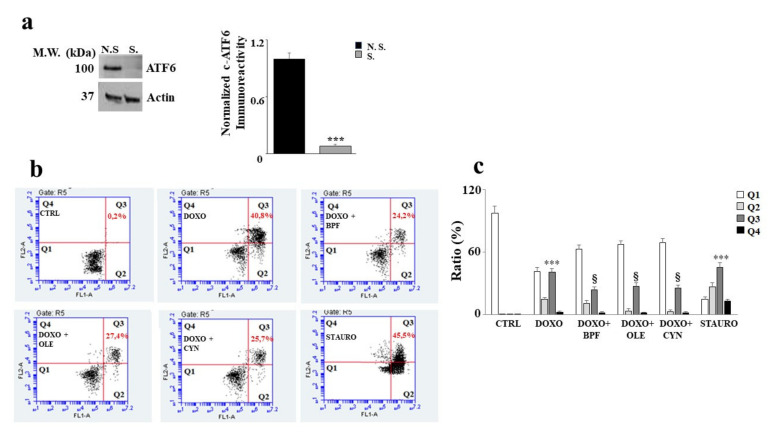
Silencing of ATF6 induces apoptotic cell death. Representative Western blot and quantification graph related to ATF6 silencing (panel **a**). N.S refers to cells that were not silenced, while S. refers to transfected and silenced cells. Representative dot plot of the cytometric analysis of three independent experiments involving Annexin V/PI staining (panel **b**). Each plot is divided into four quadrants (Q1, Q2, Q3 and Q4). Q1 refers to Annexin V negative/PI negative cells. Q2 refers to Annexin V positive/PI negative cells (apoptosis). Q3 refers to Annexin V positive/PI positive cells (apoptotic secondary Necrosis); Q4 refers to Annexin V negative/PI negative cells (advanced necrosis). Quantification graph showing mortality in cells silenced for ATF6 and treated with 1 μM of DOXO alone for 6 h, or pre-treated with BPF (5 μg/mL), OLE (1 μM), or CYN (1 μM) for 24 h and then with DOXO for 6 h (panel **c**). Cells treated with stauro-sporine (STAURO) were used as positive controls. The values of three independent experiments are expressed as means ± standard deviations. *** denotes *p* < 0.001 vs. CTRL; § denotes *p* < 0.05 vs. DOXO. The Analysis of Variance (ANOVA) was followed by the Tukey–Kramer comparison test.

**Figure 7 nutrients-13-02158-f007:**
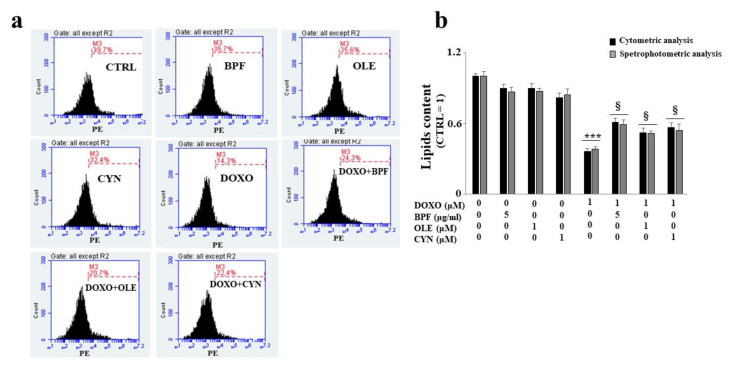
Measurement of the lipid content after ATF6 silencing. Following the silencing of the ATF6 protein, the cell lipid content was measured by cytometric and spectrophotometric analyses. A representative cytometric analysis of three independent experiments is shown; plot M3 represents a marker arbitrarily designed to show fluorescence variations (panel **a**). The relative quantification graph was constructed by setting the control value to be equal to 1 and comparing all other values to this (panel **b**). In panel b, black bars refer to the results of the cytometric analysis, while grey bars refers to the results of the spectrophotometric analysis. The values of three independent experiments are expressed as the mean ± standard deviation. *** denotes *p* < 0.001 vs. control; § denotes *p* < 0.05 vs. DOXO. The Analysis of Variance (ANOVA) was followed by the Tukey–Kramer comparison test.

**Figure 8 nutrients-13-02158-f008:**
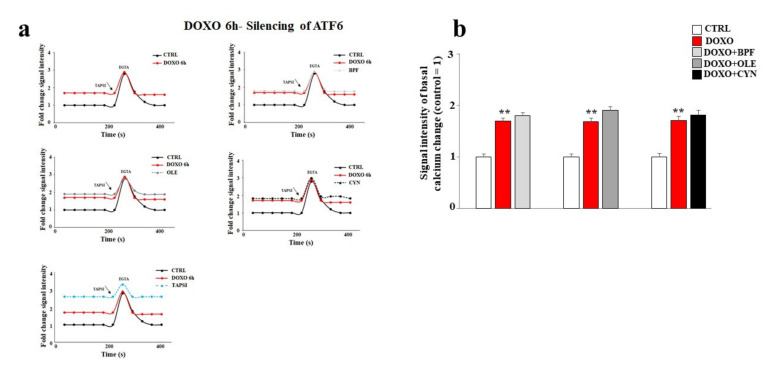
Measurement of the basal calcium ion level after ATF6 silencing. A representative graph of three independent experiments is shown (panel **a**). The basal concentration of calcium ions released from the endoplasmic reticulum was measured in cells silenced for ATF6 and treated with 1 μM DOXO alone for 6 h, or pre-treated with BPF (5 μg/mL), OLE (1 μM), or CYN (1 μM) for 24 h and then exposed to DOXO for 6 h. Cells treated with TAPSI were used as positive controls. Relative quantification of the signal intensity of the basal calcium change was conducted (panel **b**). ** denotes *p* < 0.01 vs. control. The Analysis of Variance (ANOVA) was followed by the Tukey–Kramer comparison test.

## Data Availability

The study did not report any data.

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
