# Peer review of "The Effects of Bergamot Polyphenolic Fraction, Cynara cardunculus, and Olea europea L. Extract on Doxorubicin-Induced Cardiotoxicity"

_nutrients, 2021, doi:10.3390/nu13072158_

Round 1

Reviewer 1 Report

Dear Authors,

the manuscript presents nice data, however needs to be improved before publication:

  1. The abstract should contain some summarize of the results, not only introduction.
  2. The introduction section is too long and does not explain all methods used in the experiment (why silencing of ATF6 is meeded and what information it will provide to the study.
  3. Some editorial mistakes need to be corrected (e.g. different font, two low resolution of the results).
  4. The molecular pathways should be evaluated to explain the mechanism of agents.

Author Response

Dear reviewer,

you will find, in the attached file, the answers to your requests.

Reviewer 2 Report

I was surprised by the very large number of authors. As if some were added without any input. It is not ethical.

No information on who performed the research, who analyzed it, who performed the statistical analysis? Who???

Citation: Lastname, F .; Lastname, F .; Last-name, F. Title. Nutrients 2021, 13, x. Https://doi.org/10.3390/xxxxx - it is poorly prepared, it should be corrected

Abstract is not clear for the reader and do not summarize the work. For instance: no introduction to the topic, please avoid the use of abbreviations. It should be reformulated. Add more details, emphasize the importance of the results. The authors should average this throughout the work.

The authors did not describe the statistical analysis in the text at all. This needs to be corrected.

The Material and methods section in not adequately designed. Many used techniques are not described. There is not clearly for many parameters what is their expression and measured units. This part must be strong revised and improved. In actual form it is not acceptable for publication. Add appropriate citations.

3.6. Silencing of ATF6 and its consequences - the description of this section is very poorly written. Nothing follows from this description. It is written in very plain language. This should be corrected by discussing the result in each graph

4. Discussion and conclusion - should be separated and written separately

A not good discussion of the results. The authors of the discussion discussed only a few articles. They are not new. Describe the discussion of the results in detail, recommending other recent publications.The authors did not introduce the topic. There is no such transparency, no scientific response.

No future prospects, authors should describe it.

We do not write conclusion in the form of points.

Author Response

Dear reviewer,

you will find in the attached file the answers to your requests. 

Round 2

Reviewer 1 Report

Dear Authors,

Thank you for the effort made to improve the manuscript. You still need to improve the introduction- there is too much information which not explain the idea of study e.g. there is no explanation for silencing ATF6- you need to explain why was it important and checked in this study. The introduction need to be comprised and clearly explain  the idea of the study, what is already known.

Author Response

Dear Reviewer,

The introduction was compressed but information was added about the UPR and ATF6 for purpose of clarifying the manuscript.
The overall view of the manuscript has been improved. In addition, I have sent the revised Manuscript to the service of IJMS "English editing" for a specific correction of English language.

Reviewer 2 Report

The article has been corrected.  The article should be read by the English Native Speaker. The discussion of the results is not well described. There are no conclusions that would sum up the research topic. No prospects for the future and no description of new research directions.

Author Response

Dear Reviewer,

I have sent the revised Manuscript to the service of IJMS "English editing" for a specific correction of English language.

Discussion has been revised.

Future prospectives have been added.
The overall view of the manuscript has been improved. 
